# The Influence of Manila Clam (*Ruditapes philippinarum*) on Macrobenthos Communities in a Korean Tidal Ecosystem

**Sang Lyeol Kim [1,2]**, **Hyung Gon Lee [1]**, **Su Min Kang [1]** and **Ok Hwan Yu [1,2,*]**

1   Marine Ecosystem and Biological Research Centre, KIOST, 385, Haeyang-ro, Yeongdo-gu,
    Busan 49111, Korea; boyis20c@kiost.ac.kr (S.L.K.); hglee@kiost.ac.kr (H.G.L.); kind616@kiost.ac.kr (S.M.K.)
2   Korea Maritime University, Dongsam 2-dong, Yeongdo-gu, Busan 49112, Korea
*   Correspondence: ohyu@kiost.ac.kr; Tel.: +82-051-664-3291

**Abstract:** We investigated the biological impact of extensive Manila clam (*Ruditapes philippinarum*) aquaculture on macrobenthic communities in a tidal ecosystem in Korea. We collected macrobenthos (>1 mm in length) samples seasonally in the intertidal zone in Geunsoman, Taean, Korea from April 2011 to December 2014. We identified 146 macrobenthos species, including 60 polychaetes, 53 crustaceans, and 16 mollusks. A biota–environment matching (BIO–ENV) analysis indicated that the benthic community was affected by mean sediment grain size (Mz), total organic carbon (TOC), and *R. philippinarum* biomass. We found no correlation between *R. philippinarum* and the main dominant species (*Heteromastus filiformis*, *Ceratonereis erythraeensis,* and *Ampharete arctica*), which have a different feeding strategy; thus, this may result in a lack of competition for food resources. In addition, we found that flourishing *R. philippinarum* positively affects the macrobenthos density but negatively affects the biodiversity index. Moreover, competition between species does not occur clearly, and environmental variables (sediment, organic carbon) are important.

**Keywords:** Manila clam; *Ruditapes philippinarum*; macrobenthos; feeding type; diversity

---

## 1. Introduction

The distributions of intertidal macrobenthic taxa are strongly influenced by environmental factors such as sediment type, temperature, salinity, organic carbon, etc. [1]. They play a critical role in the structure and functioning of marine ecosystems [2]. Benthos are consumed by fish and mammals, thereby providing food for higher trophic levels [3]. Macrobenthos are also important in organic matter cycling and nutrients and provide a link between the benthic and pelagic division of marine ecosystems [4]. They are used as indicators of coastal ecosystem health and environmental quality because this group is characterized by long-lived species with limited habitat ranges and high sensitivity to environmental change [5]. Macrobenthic animals are critical links between primary producers and high trophic level consumers in coastal food webs [6]. Therefore, macrobenthos are important to research targets in marine ecology and are essential to the structure and function of coastal ecosystems [7,8].

Studies of competition in marine benthic animals have been pivotal to our understanding of ecology systems overall; some of the earliest and most influential evidence of competition comes from studies of sessile marine benthic animals, and we continue to gain an understanding of community dynamics from this group [9]. Research on competition among marine invertebrates tends to focus on interference rather than exploitation [10]. Benthic organisms transport oxygen and organic matter from the surface to deeper layers, extending the habitat suitable for smaller fauna [11]. Competition, disturbance, and predation can also influence the spatial distribution of these small benthic animals [12].

Species are thought to be in fierce, direct competition for available space, where space is the primary limiting factor for marine invertebrates [13]. Evolutionary research on marine life further assumes that this competition for space is interference [14].

The Yellow Sea, situated west of Korea, is a semi-enclosed, marginal sea in the northern Pacific into which the Huanghe (Yellow) and Changjiang (Yangtze) Rivers flow. The Yellow Sea is characterized by high organic content, which originates from the surrounding landmasses. Terrigenous sediments accumulate in the subaqueous deltas of the Yellow Sea [15–17], which has resulted in the development of extensive aquaculture industry in its intertidal areas. Several bivalve species occur in the muddy and sandy tidal flats along the Yellow Sea coast in South Korea, including pacific oyster, *Magallana gigas* (Thunberg, 1793), and Manila clam, *Ruditapes philippinamm* (Adam and Reeve, 1850) [18]. *R. philippinarum* has been introduced to many parts of the world since the 1930s [19], and is now a dominant species in the intertidal zones of northwestern America, Europe, Korea, China, and Japan [20]. Manila clams are an important aquaculture species, accounting for 18% of Korea's annual shellfish production. However, after reaching a peak in 1990 (74,581 tons), *R. philippinarum* production began to decrease rapidly to 18,145 tons in 2013 and 19,853 tons in 2018 [21]. Reduced production of *R. philippinarum* may be due to a decrease in habitat area due to the reclamation of tidal flats, and mass deaths in spring and summer due to climate change [22]. Recently, Nam et al. (2018) confirmed that the parasite was closely related to the death of the clam during the high water temperature in summer [23].

*R. philippinarum* serves important biological functions in the intertidal zone, including filter-feeding, excretion, respiration, nutrient regeneration, and bio-deposition [24–27]. These processes may have an indirect impact on macrobenthos assemblages by altering the dominant species and decreasing macrobenthic fauna diversity in semi-enclosed bays [28,29]. Plankton larvae of *R. philippinarum* settle on soft substrates on reaching 0.3 mm in length, which typically occurs in spring or early autumn [30]. This bivalve grows well in an adequate mix of sand and mud (i.e., clay and silt) but tends to decline when clay and silt contents are too high. In *R. philippinarum* aquaculture, sediment composition is often deliberately altered by commercial spraying of sand or oyster shells on the seabed [31]. Attempts to increase the amount of available habitat for *R. philippinarum* should be based on an integrated analysis of environmental and biological factors related to its distribution. Water temperature, salinity, and prey species composition affect the habitat suitability for *R. philippinarum* [32,33]. Water temperature affects the duration of laying and larval growth rates; increased water temperature leads to faster clearance, ingestion, and respiration rates [34]. Although *R. philippinarum* is relatively tolerant of changes in water temperature and salinity [35], high water temperatures and low salinity can significantly affect its growth and reproduction [36–38].

Bivalves have seldom been the focus of macrobenthic community studies, although early papers assessed the impact of bivalve aquaculture on intertidal ecosystems [39,40]. In addition, most of the existing studies have been related to phytoplankton as a source of food, and there are very few articles with other benthic species in the same space. We examined the status and dynamics of intertidal macrobenthos communities by examining species richness and density, as well as community structure, in areas dominated by *Ruditapes philippinarum*. In the long term, this study will provide data to understand *R. philippinarum* management.

## 2. Materials and Methods

### 2.1. Study Area and Sampling Routine

Samples were collected seasonally (spring, summer, autumn, winter) from April 2011 to October 2014, and were conducted a total of 15 times at Geunsoman (36°43.575′ N, 126°10.269′ E), Korea (Figure 1). We collected eight replicate samples (total volume: 0.2 m$^2$, 32 core samples per year) with a can core (0.22 × 0.135 × 0.3 m) haphazardly for macrobenthos community analysis. The reasons for the single survey site were that clams do not thrive in large areas and the area (0.5 km$^2$) was limited.

Therefore, instead of multiple sites, one site was selected and sampled randomly. Samples were sieved through a 1 mm size mesh. Residue on the mesh was sorted and preserved in 10% formalin in seawater. A sample of surface sediments from the surface layer was collected to analyze sediment grain size (Mz) and total organic carbon (TOC) concentration, and these samples were frozen before analysis. In the laboratory, all organisms were sorted from the sediment and their wet weight was measured. Then, they were identified to the species level under a stereomicroscope.

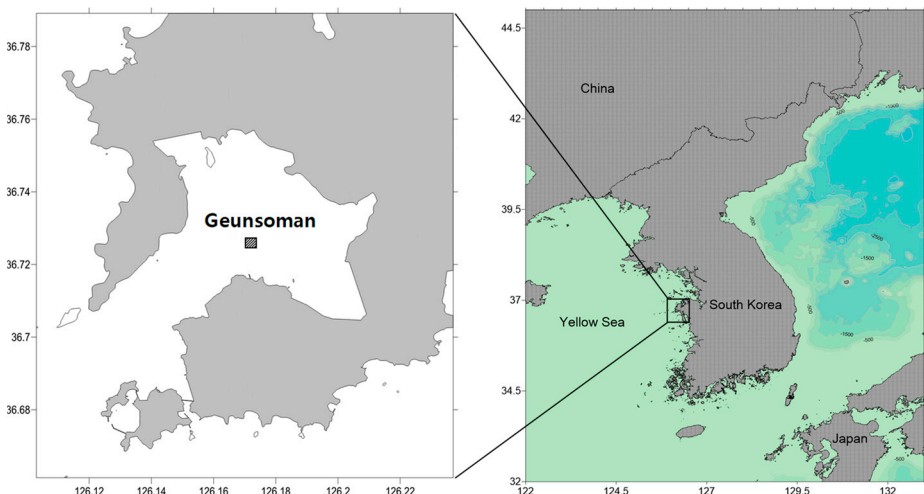

**Figure 1.** The location of sampling sites at Geunsoman, South Korea.

## 2.2. Sediment Analysis

Sediment particle sizes were determined after treating samples with a solution of 10% hydrogen peroxide. Sediment samples were heated to >100 °C to evaporate the hydrogen peroxide and then washed at least three times with distilled water to remove organisms and salts. Washed samples were then passed through a 63-μm standard sieve. After drying, the sediments trapped by the sieve were weighed and subjected to automatic particle size analysis using a SediGraph 5120 device (Kunash Instruments, Mumbai, India) following the addition of sodium hexametaphosphate as a dispersing agent. We then calculated the average particle sizes and degrees of sorting. Sediments were categorized according to Folk's classification system [41]. The content of total organic carbon (TOC) in sediments was analyzed using a Shimadzu TOC analyzer, (SSM-5000A, Shimadzu, Japan).

## 2.3. Statistical Analyses

Density and biomass data are recalculated per square meter; statistical analyses were performed for all species. The Shannon–Wiener diversity index (H') was calculated using density data. Cluster and non-metric multi-dimensional scaling (nMDS) analyses on macrobenthic community data for each sampling period were analyzed using the Bray–Curtis similarity measure based on fourth-root transformed density data and group average linkage. A similarity profile (SIMPROF) permutation test was performed to determine the statistically significant clusters among the samples. A similarity percentage (SIMPER) analysis was used to determine the contribution of each species to similarity–dissimilarity among groups. A biota–environment matching (BIO–ENV) analysis was conducted to determine the environmental factors that affect the spatial distribution of benthic animals [42]. Spearman's rank correlation analysis was used to determine relationships between biological and environmental variables. At the time of analysis, the density of *R. philippinarum* was considered an environmental variable and was not included in the calculation of Bray–Curtis similarity. The biomass of macrobenthos included all benthic animals, and only the biomass of *R. philippinarum* was calculated separately. All analyses were performed using PRIMER 6 software with the PERMANOVA add on package [43].

## 3. Results

The mean sediment grain size (Mz) was 6.46, 3.39, and 4.40 ø in April, July, and October 2011, respectively. In 2012, the average Mz was 4.75 ± 0.96 ø, which was similar to values observed in 2011 (4.75 ± 1.56 ø), but it decreased to 3.70 ± 1.56 ø in 2013 and declined further to 3.15 ± 1.25 ø in 2014 (Table 1). The TOC was 0.58, 0.31, and 0.44% in April, July, and October 2011, respectively. The average TOC was 0.99 ± 0.78% in 2012, with the highest value being recorded in October 2012 (2.15%). The average TOC was 0.48 ± 0.07% and 0.46 ± 0.09% in 2013 and 2014, respectively, similar to the values observed in 2011 (0.44 ± 0.14%) (Table 1). In the sediment type, the ratio of sand and mud was high, but overall, sand and mud were properly mixed.

**Table 1.** Sediment characteristics during the study period (Mz = mean grain size, TOC = total organic carbon).

| Year | Month | Type | Mz (ø) | TOC (%) |
|------|-------|------|--------|---------|
| 2011 | Apr | sM | 6.46 | 0.58 |
| 2011 | Jul | gmS | 3.39 | 0.31 |
| 2011 | Oct | (g)sM | 4.40 | 0.44 |
| 2012 | Jan | (g)sM | 6.15 | 0.75 |
| 2012 | Apr | (g)mS | 4.59 | 0.59 |
| 2012 | Jul | gmS | 4.16 | 0.47 |
| 2012 | Oct | gM | 4.11 | 2.15 |
| 2013 | Jan | (g)mS | 4.32 | 0.44 |
| 2013 | Apr | (g)mS | 5.40 | 0.42 |
| 2013 | Jul | gmS | 2.34 | 0.49 |
| 2013 | Oct | gmS | 2.72 | 0.58 |
| 2014 | Jan | gM | 4.34 | 0.58 |
| 2014 | Apr | zS | 4.06 | 0.42 |
| 2014 | Jul | gmS | 2.49 | 0.36 |
| 2014 | Oct | S | 1.72 | 0.47 |

Sediment type: muddy sand (mS), sand (S), gravelly sand (gS), gravel (G), sandy mud (sM), sandy gravel (sG), muddy gravel (mG), gravelly muddy sand (gmS), slightly gravelly muddy sand ((g)mS), slightly gravelly sand ((g)S), gravelly mud (gM).

We identified a total of 145 macrobenthos species. In April, July, and October 2011, 35, 45, and 47 species appeared, respectively. In 2012, the number of species detected increased from 40 in January to 55 species in April and declined to 39 in July and 36 species in October. On average, 34.8 ± 3.7 species were recorded in 2013, which increased to 40.8 ± 9.2 species in 2014 (Figure 2).

The average macrobenthos density including *Ruditapes philippinarum* was 3547 ± 2024 individuals/m$^2$ (Figure 2). In 2011, the average macrobenthos density was 3880 ± 1512 ind/m$^2$; this increased to 9470 ind/m$^2$ in January 2012, and then decreased to 4810, 5040, and 1580 ind/m$^2$ in April, July, and October 2012, respectively. The average density was 1959 ± 988 ind/m$^2$ in 2013 compared to 3580, 2950, 2905, and 3450 ind/m$^2$ in January, April, July, and October 2014, respectively. The average density of *R. philippinarum* was 1335 ± 1188 ind/m$^2$. The density was highest in January 2012 (5100 ind/m$^2$) and lowest in October 2013 (793 g/m$^2$) (Figure 2).

Throughout the study period, the average macrobenthos biomass including *Ruditapes philippinarum* was 5990 ± 3837 g/m$^2$ (Figure 2). The biomass was highest in July 2012 (13,914 g/m$^2$) and lowest in April 2014 (793 g/m$^2$). By year, 2012 was the highest (9684 ± 4511 g/m$^2$) and 2014 was the lowest (3333 ± 2960 g/m$^2$). The average *Ruditapes philippinarum* biomass was 5837 ± 3811 g/m$^2$. The highest biomass was 11,291 g/m$^2$, and the lowest biomass was 729 g/m$^2$. The *R. philippinarum* biomass was similar in shape to the macrobenthos biomass (Figure 2).

The Shannon diversity index (H') was 1.83 in April 2011, increasing to 2.31 and 2.39 in July and October 2011, respectively (Figure 2). The average H' values were relatively stable (2.25 ± 0.48 in 2012, 2.32 ± 0.12 in 2013, and 2.20 ± 0.55 in 2014). Both the species richness (d) and the Pielou's evenness (J') showed a pattern similar to the diversity index. Both indexes were highest in April 2012 (richness; 6.38, evenness; 0.73), but the richness was lowest in July 2014 (3.89) and the evenness was lowest in April 2014 (0.46).

Cluster analysis indicated two main groups (SIMPROF test, *p* < 0.001) (Figure 3). Group B was related to the October 2012, January 2013, and April 2013 sampling periods. The remaining sampling periods were all related to Group A, whereas in April 2011, it was separate from other periods. The SIMPER test indicated a 49.85% dissimilarity between Groups A and B (Table 2). A total of 15 species appeared, and of these, 6 species were polychaetes, 5 mollusks, and 4 crustaceans. The species making the largest contribution to this difference was *Ampharete arctica* Malmgren, 1866 (Polychaete), with an average dissimilarity of 1.86. The second most influential species was *Musculus senhousia* (Benson in Cantor, 1842) (Mollusca), with an average dissimilarity of 1.54. The third species was *Crangon affinis* De haan, 1849 (Crustacean), with an average dissimilarity of 1.21.

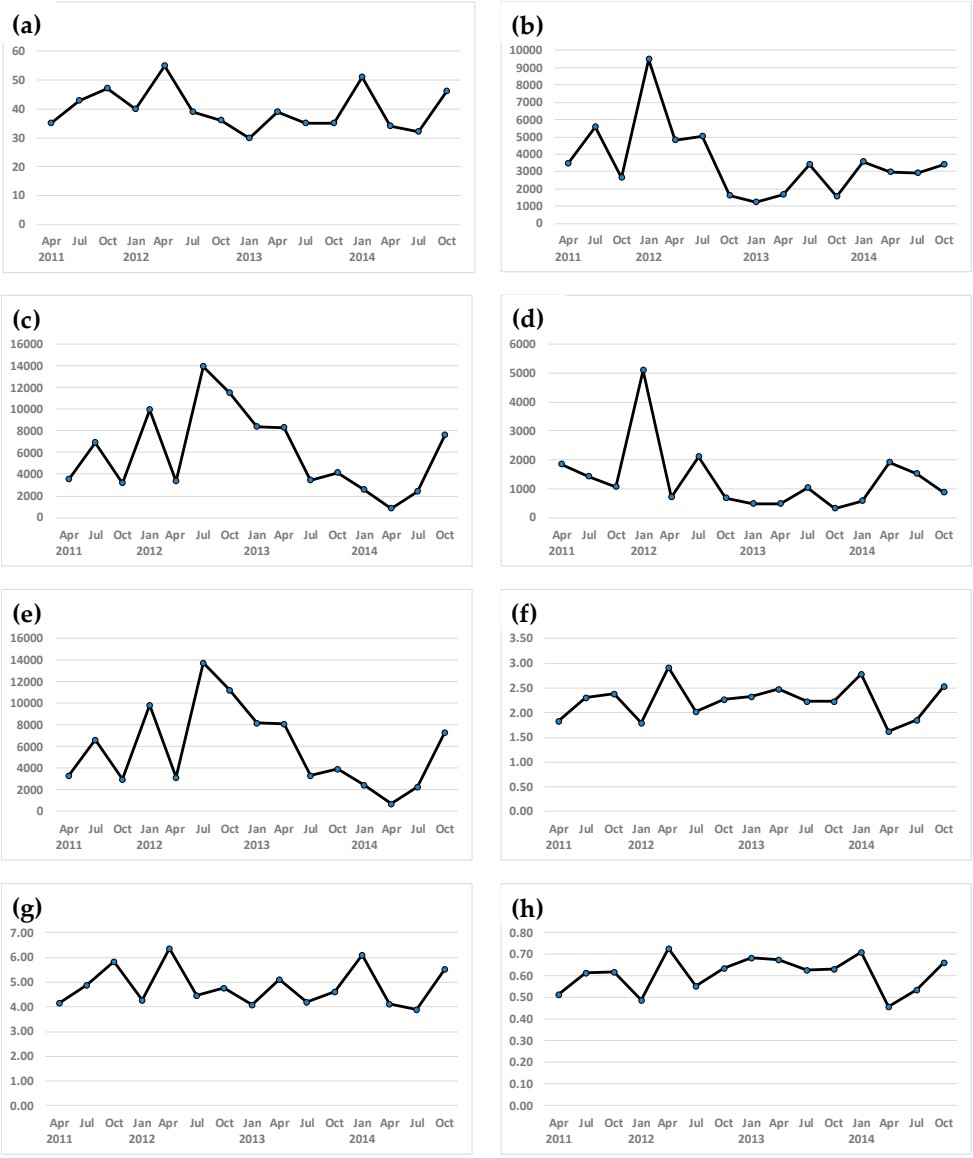

**Figure 2.** Seasonal variation of macrobenthos and ecological indices. (**a**) Macrobenthos species number, (**b**) Macrobenthos density (individuals/m$^2$), (**c**) Macrobenthos biomass (ind/m$^2$), (**d**) *Ruditapes philippinarum* density(ind/m$^2$), (**e**) *Ruditapes philippinarum* biomass (ind/m$^2$), (**f**) Shannon's diversity index: H', (**g**) Species richness: d, (**h**) Pielou's evenness: J'.

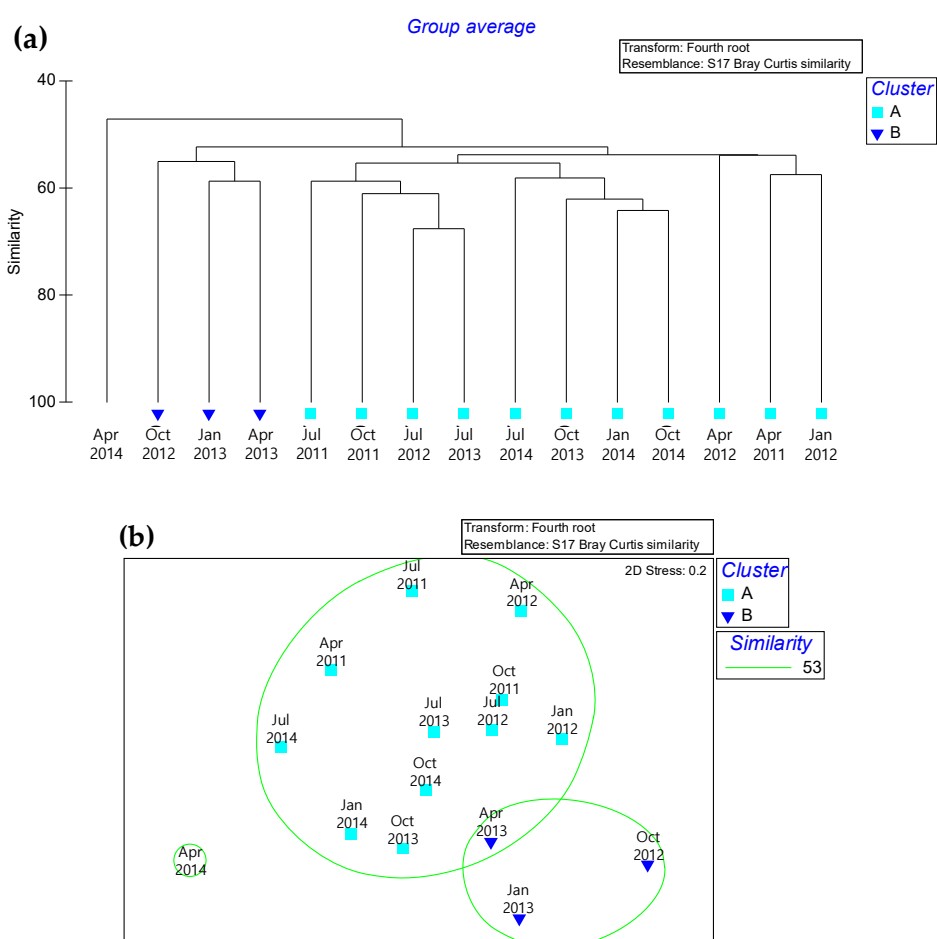

**Figure 3.** Macrobenthic community based on the Bray–Curtis similarity index. (**a**) Community cluster, (**b**) Multi-dimensional scaling (MDS) analysis.

**Table 2.** The results of similarity percentage (SIMPER) analysis and the main species in Groups A and B. The average density (fourth-root transformed data) and contribution (%) to the dissimilarity of each species are shown, as well as the cumulative percentages (P, polychaete; M, Mollusca; C, crustacean).

| Species | Group A Average Density | Group B Average Density | Average Dissimilarity | Contribution % | Cumulative % | Groups A & B Dissimilarity |
|---|---|---|---|---|---|---|
| *Ampharete arctica* (P) | 3.05 | 0 | 1.86 | 3.89 | 3.89 | 49.85 |
| *Musculus senhousia* (M) | 3.15 | 0.7 | 1.54 | 3.23 | 7.12 | |
| *Crangon affinis* (C) | 2.12 | 0.95 | 1.21 | 2.54 | 9.67 | |
| *Chone teres* (P) | 1.85 | 0 | 1.1 | 2.3 | 11.97 | |
| *Ilyoplax pingi* (C) | 1.87 | 1.2 | 0.85 | 1.77 | 13.74 | |
| *Amaeana occidentalis* (P) | 1.32 | 0 | 0.78 | 1.64 | 15.38 | |
| *Diastylis paratricincta* (C) | 1.8 | 1.47 | 0.78 | 1.63 | 17.01 | |
| *Mediomastus californiensis* (P) | 1.51 | 0.5 | 0.77 | 1.61 | 18.63 | |
| *Ruditapes philippinarum* (M) | 5.93 | 4.79 | 0.76 | 1.59 | 20.22 | |
| *Philine argentata* (M) | 1.16 | 0.81 | 0.75 | 1.56 | 21.78 | |
| *Palaemon serrifer* (C) | 0 | 1.19 | 0.73 | 1.53 | 23.31 | |
| *Grandidierella japonica* (C) | 2.51 | 1.34 | 0.72 | 1.52 | 24.83 | |
| *Reticunassa festiva* (M) | 1.05 | 1.9 | 0.65 | 1.37 | 26.2 | |
| *Anoides oxycephala* (P) | 1.31 | 0.5 | 0.65 | 1.37 | 27.56 | |
| *Eteone longa* (P) | 1.42 | 1.2 | 0.61 | 1.28 | 28.84 | |

The BIO–ENV analyses and nMDS bubble plot showed the density and biomass of *Ruditapes philippinarum*, and the TOC showed the highest correlations with a macrofaunal composition (Rho = 0.32, $p < 0.05$; Table 4). However, the mean grain size (Mz) had relatively little effect (Table 3, Figure 4).

**Table 3.** Environmental and biological variables affecting the macrobenthos community as determined by a biota–environment matching (BIO–ENV) analysis (Mz = mean grain size, TOC = total organic carbon, RP = *Ruditapes philippinarum*).

| Number of Variables | Correlation (%) | Best Variables |
|:---:|:---:|:---|
| 3 | 0.320 | TOC, RP density, RP biomass |
| 2 | 0.299 | TOC, RP biomass |
| 2 | 0.289 | TOC, RP density |
| 1 | 0.270 | TOC |
| 3 | 0.251 | Mz, TOC, RP biomass |
| 1 | 0.249 | RP biomass |
| 4 | 0.247 | Mz, TOC, BP density, RP biomass |
| 2 | 0.244 | BP density, RP biomass |
| 2 | 0.208 | Mz, TOC |
| 3 | 0.199 | Mz, TOC, BP density |

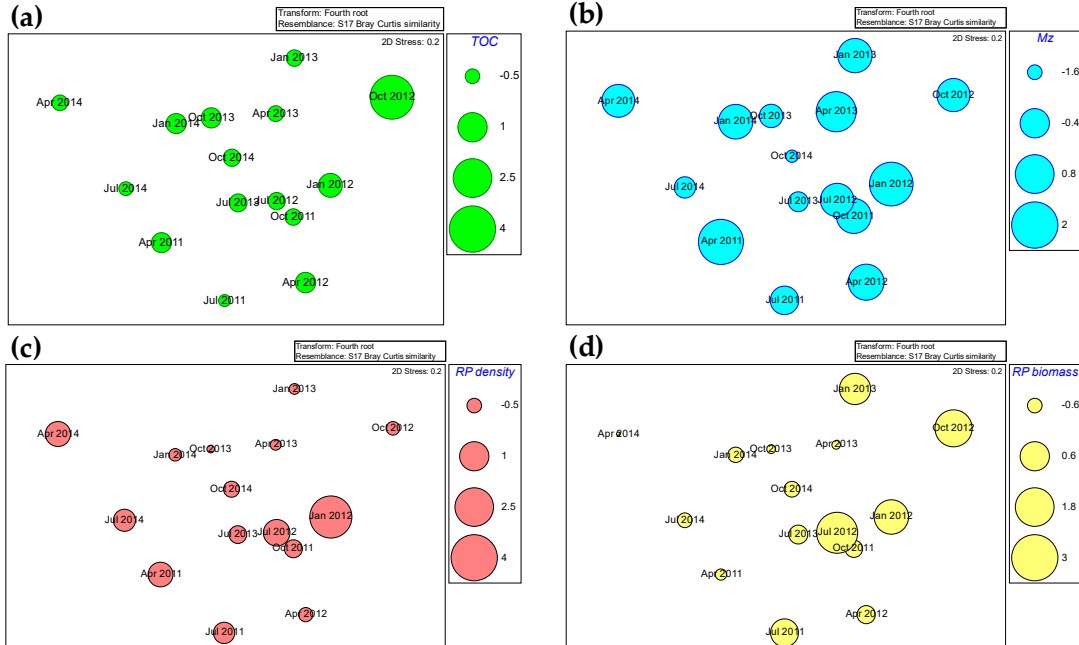

**Figure 4.** Non-metric multidimensional scaling (nMDS) bubble plot of environmental and biological variables. (**a**) Total organic carbon (TOC), (**b**) Mean grain size (Mz), (**c**) *Ruditapes philippinarum* (RP) density, (**d**) *Ruditapes philippinarum* (RP) biomass.

Correlation analysis showed that the number of macrobenthos species was positively correlated with the Shannon diversity index (Table 4). The diversity index had a negative correlation with *Ruditapes philippinarum* density. Additionally, macrobenthos biomass including *R. philippinarum* was positively correlated with *R. philippinarum* biomass. Two dominant species (*Heteromastus filiformis* and *Ceratonereis erythraeensis*) were positively correlated with the number of macrobenthos species; these two species are deposit and detritus feeders polychaetes, respectively, whereas *R. philippinarum* is a suspension feeder.

**Table 4.** Spearman rank correlations between biological and environmental variables (Mz, TOC, macrobenthos species richness, biomass, and density, Shannon diversity: H′, *Ruditapas philippinarum* (RP) density and biomass) (HF = *Heteromastus filiformis*, CE = *Ceratonereis erythraeensis*, AA = *Ampharete arctica*) (* = $p < 0.05$; ** = $p < 0.01$, *** = $p < 0.001$).

| | TOC | Species | Density | Biomass | H′ | RP$^{Su}$ den | RP$^{Su}$ bio | HF$^{De}$ | CE$^{Dt}$ | AA$^{De}$ |
|---|---|---|---|---|---|---|---|---|---|---|
| Mz | 0.320 | 0.271 | 0.211 | 0.146 | 0.043 | 0.118 | 0.100 | 0.004 | 0.182 | −0.460 |
| TOC | | 0.271 | 0.144 | 0.286 | 0.050 | −0.115 | 0.279 | 0.273 | 0.363 | −0.374 |
| Species | | | 0.515 * | 0.004 | 0.648 ** | −0.020 | 0.004 | 0.608 * | 0.705 ** | 0.133 |
| Density | | | | 0.021 | −0.125 | 0.657 ** | 0.018 | 0.493 | 0.714 ** | 0.453 |
| Biomass | | | | | −0.004 | −0.054 | 0.996 *** | 0.339 | 0.300 | −0.357 |
| H′ | | | | | | −0.679 ** | 0.007 | 0.232 | 0.289 | −0.072 |
| RP$^{Su}$ den | | | | | | | −0.068 | −0.004 | 0.179 | 0.422 |
| RP$^{Su}$ bio | | | | | | | | 0.346 | 0.318 | −0.327 |
| HF$^{De}$ | | | | | | | | | 0.779 *** | −0.027 |
| CE$^{Dt}$ | | | | | | | | | | 0.164 |
| AA$^{De}$ | | | | | | | | | | |

(Su; suspension feeder, De; deposit feeder, Dt; detritus feeder).

## 4. Discussion

*Ruditapes philippinarum* accounted for 37.6% of the total macrobenthos density and had greater biomass than all other species. Therefore, this species occupied more space than the other species in the study area and led to reduced macrobenthos species richness. The use of univariate measures of diversity (Shannon–Weaver diversity; H′) is used to assess the level of stress in macrobenthos communities [44,45]. The index (H′) was highly related to macrobenthos species richness [46]. Somerfield et al. (2009) [47] suggested that local species diversity is determined by disturbance, predation, and competition, and also by environmental structure and regional processes. In this study, the prosperity of the clams made the diversity index low. This is because a large increase in one species affects the entire benthic ecosystem. The correlation between the clams and other dominant species was not significantly related. This is because they do not compete for food, but coexist and survive.

Spatial patterns of dominant species allow us to understand the structure of target populations [48,49]. Choi (2003) [50] found that benthic groups at Gwangyang reflected the degree of dominance and regional distribution of dominant species. The proportion of the community occupied by dominant species plays a significant role in the overall community structure and provides a lens through which to interpret environmental conditions [51,52]. Although *Ruditapes philippinarum* was affected by multiple environmental factors (i.e., sediment type, temperature, salinity, organic carbon, etc.), it was most influenced by sediment composition [53]. Sediment characters are the key drivers of macrobenthos communities [54]. The proportion of sand in the substrate is important for *Ruditapes philippinarum*, which prefers substrates with a 50%–80% sand content [55]. The sediment in our study area had a sand content of 54%, which is suitable for *R. philippinarum*. Previous research in four areas in Gyeonggi Bay that had high densities of *R. philippinarum* found an average sediment grain size of 3.8 ± 0.1 ø [56], which was similar to the average grain size in our study area (4.1 ± 1.4 ø). The grain size and composition of sediment affect the lifecycle of benthic animals [57]. In a study conducted in Seonjaedo, Korea, Kim (2005) [58] reported an Mz of 3.47 ± 0.45 ø in the eastern part of the study area and 3.60 ± 0.34 ø in the western part. Thus, our study area had a sediment environment suitable for the clams to thrive and also had appropriate substrate conditions for clam growth.

A biota–environment matching (BIO–ENV) analysis indicated that the total organic carbon (TOC) influenced benthos communities. Analyzing environmental variables such as TOC is important for evaluating coastal marine ecosystems [59]. TOC is highly related to benthic food sources and is therefore associated with macrobenthic fauna [60]. Because the amount of clams is overwhelming, their presence has influenced the benthic fauna community, and it is necessary to examine the relationship with the dominant species other than the clams. The second most dominant species, the polychaete *Heteromastus filiformis*, is also a dominant species in other areas, such as Asan Bay and Gyeonggi Bay [61].

Spiridonov (2016) [62] suggested that *H. filiformis* prefer fine-grained sediments with high organic matter content. *H. filiformis* and the other dominant species, *Ceratonereis erythraeensis*, are deposit and detritus feeders, respectively, whereas *R. philippinarum* is a suspension feeder [63]. Deposit feeders consume organic matter attached to sediment particles, whereas detritus feeders obtain nutrients by consuming detritus [64]. Due to these major differences in feeding type, there would be little competition between *R. philippinarum* and these two dominant species.

We recorded 145 macrobenthos species at Geunsoman, similar to Antoniadou (2010) and findings from Anmyeon Island [65,66]. Furthermore, macrobenthos species richness can differ according to survey timing and sampling methods. However, broad patterns of diversity in similar areas can be predicted with reasonable accuracy [67]. In particular, polychaetes are in an important position. Polychaetes alter the quality and size of sedimentary facies through feeding and therefore play an important role in benthic ecosystems [68,69]. In this study, polychaete species comprised 55% of the total macrobenthos population; the respective values were 67% for the Hallyeohaesang National Park area, 56% for the Kakinada tidal flat, and 50% for the Mormugao tidal flat [70,71]. We found no correlation of *Ruditapes philippinarum* with overall species richness, nor with the richness or proportion of polychaetes species. The correlation analysis indicated that *R. philippinarum* was not associated with other dominant species.

**Author Contributions:** Conceptualization, S.L.K. and O.H.Y.; formal analysis, S.L.K., S.M.K., H.G.L. and O.H.Y.; project administration, O.H.Y.; writing—original draft, S.L.K. and O.H.Y. All authors have read and agreed to the published version of the manuscript.

**Funding:** This research was supported by the project "Oil spill environmental impact assessment and environmental restoration (PM57431)" funded by the Ministry of Oceans and Fisheries of Korea. This work was also supported by the project "A base study of understanding and counteract marine ecosystem change in Korean waters (PE99813)" funded by the Korea Institute of Ocean Science and Technology.

**Conflicts of Interest:** The authors declare no conflict of interest.

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
