# Peer review of "The Influence of Manila Clam (Ruditapes philippinarum) on Macrobenthos Communities in a Korean Tidal Ecosystem"

_sustainability, doi:10.3390/su12104205_

Round 1

Reviewer 1 Report

Most suggestions by reviewers were accepted. The paper has markedly improved, though I still have some doubts about the sampling design. However, it seems now acceptable for publication. 

I only made a couple of minor revision in the enclosed word file.

Author Response

We have revised the manuscript in line with the suggestions made by one reviewer and accepted most of suggestions. Here we enclosed a written answer on the comments of the reviewer.

Reviewers’ comments:

Reviewer 1

"As a matter of fact there was only 'Philippinarum' to correct to 'philippinarum' in the title. The rest was correct. I appreciated the authors emended most of our coments."

à As your comments, we changed the sentence to 'philippinarum'.

àL 2-4 The Influence of Manila clam (Ruditapes philippinarum) on Macrobenthos Communities in a Korean Tidal Ecosystem

Reviewer 2 Report

The paper describes a macrofaunal community in association with R. philippinarum.  I have some concerns and recommendation related to the design and statistical analyses of this study.  Specific comments are listed below.

Introduction

This section mostly reads fine.  There are some repetitive sentences that can be organized for a better flow.  The last 2 paragraphs seem to jump around a bit (though I can sort of guess how they can be connected) and should be be re-written to emphasize how benthic fauna interacts with one another and why your study is important.

The introduction section should first start with a broad idea then be narrowed down to topics specific to your study.  In this sense, I recommend moving the 2 paragraphs from Line 55 to Line 75 to the beginning of the section, then you can discuss the specific geographical location (i.e. the Yellow Sea) and R. philippinarum.  In this way, it should be easier to discuss why it was important to investigate your specific case.

You state that R. philippinarum production has declined, yet you are investigating its influence on macrobenthic communities.  Are there any environmental concerns regarding the R. philippinarum aquaculture despite the declining productivity?  This is the type of sentences that need to be stated explicitly, so that readers who are not familiar with the R. philippinarum aquaculture in Korea can actually understand why your study is important and meaningful.

L63. "Therefore, macrobenthos is important to research targets in marine ecology"  Do you mean "macrobenthos are important research targets in marine ecology"?

Materials and Methods

Lines 86-89. So, is "Geunsoman, Sowon-myeon, Taean-gun, and Chungcheongnam-do, Korea" actually a 1 site?  It sounds like 4 different sites in Korea, and the map (Fig 1) only shows Geunsoman. I have no idea how important it is to include "Sowon-myeon, Taean-gun, and Chungcheongnam-do" in the main text, but would it be a problem to just say "Geunsoman, Korea" as it is much less confusing?  Could you also clarify that 8 replicate core samples were collected from Geunsoman per season, so 32 sediment cores per year?

Lines 89-98. Could you clarify how did you randomized your sampling?  Please also add the spatial area (size, e.g. 1 km2) of your site from which random samples were taken.

Line 111. You used cluster analysis and SIMPROF test.  nMDS is not a test.  It is simply a data visualization tool.

Lines 115-120. These sentences are confusing.  It says "The pattern of environmental variables was ordinated using the principal component analysis (PCA)" and "The PCA was performed on macrobenthos groups".  Which one is it?  It is generally not appropriate to use PCA with biological data, so it needs to be thoroughly justified if PCA was performed on macrobenthos.  In addition, looking at Figure 4, neither was actually done.  What authors did was PCO on the basis of the Bray-Curtis similarity.

Results

Table 1. Please clarify what those parentheses mean in the Type column.

Lines 147-152. How was biomass calculated?  This didn't seem to be explained in the method section.

Line 162.  Why were the 8 replicates pooled for the clustering/SIMPROF tests?  The tests should have been done at the replicate level or the pooling process needs to be explained and thoroughly justified in the method section.

Lines 165-166.  I do not understand what you mean by "A total of 15 species were less than 30% cumulative".  Please rephrase.

Figure 3. There is no point of showing both cluster tree and nMDS.  Please pick 1, either the cluster tree or nMDS showing the grouping identified by the cluster analysis.

Line 179. What is shown in Fig 4 is not a result of PCA.  It says PCO and looking at the figure caption it confirms it is PCO.  PCA is inherently based on the Euclidean distance and it cannot be done on the basis of Bray-Curtis.  In addition, give the low variability captured by the PCO ordination (only 32% of the variability present in the macrobenthic community data), what is the point of the vector overlay showing correlations with some environmental variables?  The PCO here is clearly not a good visualization of the similarity of macrobenthic community, given Apr 2013 data is on top of those from Cluster A.  If you feel strongly about visualizing the result of the BIO-ENV test, I recommend using a bubble plot on nMDS.

Table 4. Was the density of Ruditapes philippinarum included in the calculation of the Bray-Curtis similarity?  Please clarify this in the method.  If their density is considered an environmental variable, it should not be included in the calculation of the Bray-Curtis similarity.  In terms of looking at the community structure as a whole, however, they should be included in the calculation of the Bray-Curtis similarity and should not be used as an environmental variable.

Did you also check for a correlation between RP density and RP biomass?  If they are highly correlated, you should not be using both as predictor variables at the same time.

Lines 190-191. "Additionally, macrobenthos biomass was positively correlated with R. philippinarum biomass."  Again, was the R. philippinarum biomass included in the macrobenthos biomass?  Please clarify this in the method section.

Discussion

This section is a bit confusing as I cannot separate what are the results of your study vs. what have been already published.  Please go over the section and make sure the distinction is clearly made throughout it.

Lines 208-209. "The correlation between the clams and other species was not significantly related to other species. In other words, no animal damaged the clam’s habitat."  I do not understand what these sentences are saying.  Please rephrase.

Lines 211-227. Was grain size or %sand etc included in the BIO-ENV analysis?

Author Response

We have revised the manuscript in line with the suggestions made by the reviewer and accepted most of suggestions. Here we enclosed a written answer on the comments of the reviewer.

Reviewer 2

  1. This section mostly reads fine. There are some repetitive sentences that can be organized for a better flow. The last 2 paragraphs seem to jump around a bit (though I can sort of guess how they can be connected) and should be be re-written to emphasize how benthic fauna interacts with one another and why your study is important. The introduction section should first start with a broad idea then be narrowed down to topics specific to your study. In this sense, I recommend moving the 2 paragraphs from Line 55 to Line 75 to the beginning of the section, then you can discuss the specific geographical location (i.e. the Yellow Sea) and R. philippinarum. In this way, it should be easier to discuss why it was important to investigate your specific case.

à As your comments, we moved the last 2 paragraphs to the beginning of the section.

àL 26-36 The distributions of intertidal macrobenthic taxa are strongly influenced by environmental factors such as sediment type, temperature, salinity, organic carbon, etc. [1]. They play a critical role in the structure and functioning of marine ecosystems [2]. Benthos are consumed by fish, and mammals, thereby providing food for higher trophic levels [3]. Macrobenthos is also important in organic matter cycling and nutrient and provides a link between the benthic and pelagic division of marine ecosystems [4]. They are used as indicators of coastal ecosystem health and environmental quality because this group is characterized by long-lived species with limited habitat ranges and high sensitivity to environmental change [5]. Macrobenthic animals are critical links between primary producers and high trophic level consumers in coastal food webs [6]. Therefore, macrobenthos are important research targets in marine ecology that are essential to the structure and function of coastal ecosystems [7,8].

  1. You state that R. philippinarum production has declined, yet you are investigating its influence on macrobenthic communities.  Are there any environmental concerns regarding the R. philippinarumaquaculture despite the declining productivity?  This is the type of sentences that need to be stated explicitly, so that readers who are not familiar with the R. philippinarum aquaculture in Korea can actually understand why your study is important and meaningful.

àAs your comments, we added the sentence to “Reduced production of R. philippinarum may be due to a decrease in habitat area due to reclamarion of tidal flats, and mass deaths in spring and summer due to climate change [23]. Recently, Nam et al. (2018) confirmed that the parasite was closely related to the death of the clam during the high water temperature in summer [24].”.

àL 59-62 Reduced production of R. philippinarum may be due to a decrese in habitat area due to reclamarion of tidal flats, and mass deaths in spring and summer due to climate change [23]. Recently, Nam et al. (2018) confirmed that the parasite was closely related to the death of the clam during the high water temperature in summer [24]

  1. L63. "Therefore, macrobenthos is important to research targets in marine ecology” Do you mean "macrobenthos are important research targets in marine ecology"?

àAs your comments, we changed the sentence to “macrobenthos are important research targests in marine ecology.”.

àL 34-36 Therefore, macrobenthos are important research targets in marine ecology that are essential to the structure and function of coastal ecosystems [7,8].

  1. Lines 86-89. So, is "Geunsoman, Sowon-myeon, Taean-gun, and Chungcheongnam-do, Korea" actually a 1 site?  It sounds like 4 different sites in Korea, and the map (Fig 1) only shows Geunsoman. I have no idea how important it is to include "Sowon-myeon, Taean-gun, and Chungcheongnam-do" in the main text, but would it be a problem to just say "Geunsoman, Korea" as it is much less confusing?

à As your comments, we changed the sentence to “and were conducted a total of 15 times at Geunsoman (36°43.575′N, 126°10.269′E).”.

àL 87-89 Samples were collected seasonally (spring, summer, autumn, winter) from April 2011 to October 2014, and were conducted a total of 15 times at Geunsoman (36°43.575′N, 126°10.269′E), Korea (Figure 1).

  1. Could you also clarify that 8 replicate core samples were collected from Geunsoman per season, so 32 sediment cores per year?

à As your comments, we changed the sentence to “We collected in eight replicate samples (total volume: 0.2 m2, 32 core samples per year)”.

àL 89 We collected in eight replicate samples (total volume: 0.2 m2, 32 core samples per year)

  1. Lines 89-98. Could you clarify how did you randomized your sampling?  Please also add the spatial area (size, e.g. 1 km2) of your site from which random samples were taken.

à As your comments, we added the spatial area as “the area (0.5 km2)”

àL 90-91 The reason for the single survey site was that clams did not thrive in large areas and the area (0.5 km2) was limited.

  1. Line 111. You used cluster analysis and SIMPROF test.  nMDS is not a test.  It is simply a data visualization tool.

à As your comments, we changed the sentence to “A similarity profile (SIMPROF) permutation tests were performed to determine the statistically significant clusters among samples.”.

àL 111-118 Cluster and non-metric multi-dimensional scaling (nMDS) analyses on macrobenthic community data for each sampling period were analyzed using the Bray-Curtis similarity measure based fourth-root transformed density data and group average linkage. A similarity profile (SIMPROF) permutation tests were performed to determine the statistically significant clusters among samples. A similarity percentage (SIMPER) analysis was used to determine the contribution of each species to similarity-dissimilarity among groups

  1. Lines 115-120. These sentences are confusing.  It says "The pattern of environmental variables was ordinated using the principal component analysis (PCA)" and "The PCA was performed on macrobenthos groups".  Which one is it?  It is generally not appropriate to use PCA with biological data, so it needs to be thoroughly justified if PCA was performed on macrobenthos.  In addition, looking at Figure 4, neither was actually done.  What authors did was PCO on the basis of the Bray-Curtis similarity.

àAs your comments, we deleted the paragraphs relate to PCA.

  1. Table 1. Please clarify what those parentheses mean in the Type column.

à As your comments, we changed the sentence to “Sediment type; muddy sand [mS], sand [S], gravelly sand [gS], gravel [G], sandy mud [sM], sandy gravel [sG], muddy gravel [mG], gravelly muddy sand [gmS], slightly gravelly muddy sand [(g)mS], slightly gravelly sand [(g)S], gravelly mud [gM].”.

àL 137-139 Sediment type; muddy sand [mS], sand [S], gravelly sand [gS], gravel [G], sandy mud [sM], sandy gravel [sG], muddy gravel [mG], gravelly muddy sand [gmS], slightly gravelly muddy sand [(g)mS], slightly gravelly sand [(g)S], gravelly mud [gM].

  1. Lines 147-152. How was biomass calculated?  This didn't seem to be explained in the method section.

à As your comments, we added the sentence to “all organisms were sorted from the sediment and measured wet weight. Then, they were identified to the species level under a stereomicroscope”.

àL 95-97 In the laboratory, all organisms were sorted from the sediment and measured wet weight. Then, they were identified to the species level under a stereomicroscope.

  1. Line 162.  Why were the 8 replicates pooled for the clustering/SIMPROF tests?  The tests should have been done at the replicate level or the pooling process needs to be explained and thoroughly justified in the method section.

à The 8 replicateds was associated with a total volume. The 0.2 m2 volume was considered suitable. We added the sentence to “We collected in eight replicate samples (total volume: 0.2 m2, 32 core samples per year).”.

àL 89-90 We collected in eight replicate samples (total volume: 0.2 m2, 32 core samples per year) randomly for macrobenthos community analysis.

  1. Lines 165-166.  I do not understand what you mean by "A total of 15 species were less than 30% cumulative".  Please rephrase.

à As your comments, we changed the sentence to “A total of 15 species appeared, of these, 6 species were polychaetes, 5 mollusks, and 4 crustaceans.”.

àL 169-170 A total of 15 species appeared, of these, 6 species were polychaetes, 5 mollusks, and 4 crustaceans.

  1. Figure 3. There is no point of showing both cluster tree and nMDS.  Please pick 1, either the cluster tree or nMDS showing the grouping identified by the cluster analysis.

à As your comments, we added to nMDS similarity bubbles.

àL 178-179 Figure 3. Clustering and MDS analysis of the macrobenthic community based on the Bray-Curtis similarity index.

  1. Line 179. What is shown in Fig 4 is not a result of PCA.  It says PCO and looking at the figure caption it confirms it is PCO.  PCA is inherently based on the Euclidean distance and it cannot be done on the basis of Bray-Curtis.  In addition, give the low variability captured by the PCO ordination (only 32% of the variability present in the macrobenthic community data), what is the point of the vector overlay showing correlations with some environmental variables?  The PCO here is clearly not a good visualization of the similarity of macrobenthic community, given Apr 2013 data is on top of those from Cluster A.  If you feel strongly about visualizing the result of the BIO-ENV test, I recommend using a bubble plot on nMDS.

à As your comments, we deleted contents related to PCA and added to nMDS bubbles plot.

àL 190-191 Figure 4 Non-metric Multidimensional Scaling (nMDS) bubble plot of environmental and biological variables (Mz = mean grain size, TOC = total organic carbon, RP = Ruditapes philippinarum).

  1. Table 4. Was the density of Ruditapes philippinarum included in the calculation of the Bray-Curtis similarity?  Please clarify this in the method.  If their density is considered an environmental variable, it should not be included in the calculation of the Bray-Curtis similarity.  In terms of looking at the community structure as a whole, however, they should be included in the calculation of the Bray-Curtis similarity and should not be used as an environmental variable.

à As your comments, we added the sentence to “At the time of analysis, the density of benthic animals was considered an environmental variable and was not included in the calculation of Bray-Curtis similarity.”.

àL 121-122 At the time of analysis, the density of benthic animals was considered an environmental variable and was not included in the calculation of Bray-Curtis similarity. ].

  1. Did you also check for a correlation between RP density and RP biomass?  If they are highly correlated, you should not be using both as predictor variables at the same time.

à As a result of correlation analysis, there was no relationship between density and biomass of R. philippinarum.

  1. Lines 190-191. "Additionally, macrobenthos biomass was positively correlated with R. philippinarum biomass."  Again, was the R. philippinarum biomass included in the macrobenthos biomass?  Please clarify this in the method section.

à As your comments, we added the sentence to “The biomass of macrobenthos included all benthic animals, and only the biomass of R. philippinarum was calculated separately.”.

àL 123-124 The biomass of macrobenthos included all benthic animals, and only the biomass of R. philippinarum was calculated separately.

  1. This section is a bit confusing as I cannot separate what are the results of your study vs. what have been already published.  Please go over the section and make sure the distinction is clearly made throughout it.

à As your comments, we changed several sentences.

àL 200-203 Therefore, this species occupies more space than the other species in the study area and has led to reduced macrobenthos species richness. The use of univariate measures of diversity (Shannon-Weaver diversity; H’) to evaluate the level of stress- induced on macrobenthos communities [44,45].

àL 217-218 Choi (2003) [50] found that benthic groups at Gwangyang reflected the degree of dominance and regional distribution of dominant species.

àL 232-233 A biota–environment matching (BIO-ENV) analysis indicated that the total organic carbon (TOC), influenced benthos communities. ].

  1. Lines 208-209. "The correlation between the clams and other species was not significantly related to other species. In other words, no animal damaged the clam’s habitat."  I do not understand what these sentences are saying.  Please rephrase.

à As your comments, we changed the sentence to “The correlation between the clams and other dominant species was not significantly related.”.

àL 207-208 The correlation between the clams and other dominant species was not significantly related.

  1. Lines 211-227. Was grain size or %sand etc included in the BIO-ENV analysis?

à The mean grain size shows the sediment type according to the value. Therefore, other sediment elements such as sand ratio were not included.

We believe the manuscript has been improved satisfactorily and hope it will be revised for publication in Special Issue “Harmful organisms and their management for sustainable environment” of Sustainability.

Best regards,

OK HWAN YU

Round 2

Reviewer 2 Report

The manuscript is in a much better shape.

Here are some minor comments.

L90. Please briefly explain how the randomization was achieved (e.g. generating a grid system for the study area and using a random number generator to find 8 grids for sampling, etc). If the sampling was not truly randomized, it should say "haphazardly" instead.

L121-124. "At the time of analysis, the density of benthic animals was considered an environmental variable and was not included in the calculation of Bray-Curtis similarity." This sentence does not make sense. If the density of benthic animals was NOT included in the calculation of Bray-Curtis similarity, what was included in the calculation? It says in L114 "the Bray-Curtis similarity measure based fourth-root transformed density data."  I hope it was meant to say "the density of R. philippinarum."

L144. "The average macrobenthos density was 3,547 ± 2,024 individuals / m2 (Figure 2)." Is this including R. philippinarum or excluding R. philippinarum? Please clarify in the text. It seems in Figure 2 that macrobenthos dentsity and R. philippinarum density are highly correlated, so it makes me wonder if R. philippinarum density was included in the calculation of macrobenthos dentsity, and it would be very helpful if it is explicitly stated here.

L151. "the average macrobenthos biomass was 5,990 ± 3,837 g/m2 (Figure. 2)." Again, is this including R. philippinarum or excluding R. philippinarum?  Please clarify in the text.

L181. Please change the word "abundance" to "density" for the sake of consistency. The same goes to the column labels so they read "average density."

L194. "Additionally, macrobenthos biomass was positively correlated with R. philippinarum biomass." Again, please clarify whether R. philippinarum biomass was included in the calculation of macrobenthos biomass in the text. This is important, as if it was NOT included, it means that higher biomass of R. philippinarum is associated with higher biomass of other organisms, which does not seem to make sense given the negative correlation between R. philippinarm and Shannon index, but I guess it can happen... If it was included, it probably simply means that a relatively large proportion of the macrobenthos biomass is the biomass of R. philippinarum.

L201-202. "The use of univariate measures of diversity (Shannon-Weaver diversity; H’) to evaluate the level of stress-induced on macrobenthos communities [44,45]." This is not a complete sentence. Please rephrase.

Author Response

Reviewers’ comments:

Reviewer 2

  1. L90. Please briefly explain how the randomization was achieved (e.g. generating a grid system for the study area and using a random number generator to find 8 grids for sampling, etc). If the sampling was not truly randomized, it should say "haphazardly" instead.

à As your comments, we changed the sentence “We collected in eight replicate samples (total volume: 0.2 m2, 32 core samples per year) with a can core (0.22 x 0.135 x 0.3 m) haphazardly for macrobenthos community analysis.”

àL 89-90 We collected in eight replicate samples (total volume: 0.2 m2, 32 core samples per year) with a can core (0.22 x 0.135 x 0.3 m) haphazardly for macrobenthos community analysis

  1. L121-124. "At the time of analysis, the density of benthic animals was considered anenvironmental variable and was not included in the calculation of Bray-Curtis similarity." This sentence does not make sense. If the density of benthic animals was NOT included in the calculation of Bray-Curtis similarity, what was included in the calculation? It says in L114 "the Bray-Curtis similarity measure based fourth-roottransformed density data."  I hope it was meant to say "the density of R. philippinarum."

à As your comments, we changed the sentence as “At the time of analysis, the density of R. philippinarum was considered an environmental variable and was not included in the calculation of Bray-Curtis similarity “.

àL 121-124 At the time of analysis, the density of R. philippinarum was considered an environmental variable and was not included in the calculation of Bray-Curtis similarity

  1. L144. "The average macrobenthos density was 3,547 ± 2,024 individuals / m2 (Figure 2)." Is this including R. philippinarum or excluding R. philippinarum? Please clarify in the text. It seems in Figure 2 that macrobenthos dentsity and R. philippinarum density are highly correlated, so it makes me wonder if R. philippinarum density was included in the calculation of macrobenthos dentsity, and it would be very helpful if it is explicitly stated here.

à As your comments, we added the sentence “including Ruditapes philippinarum”.

àL 144 The average macrobenthos density including R. philippinarum was 3,547 ± 2,024 individuals / m2 (Figure 2).

  1. L151. "the average macrobenthos biomass was 5,990 ± 3,837 g/m2 (Figure.2)." Again, is this including R. philippinarum or excluding R. philippinarum?  Please clarify in the text.

à As your comments, we added the sentence “including Ruditapes philippinarum”.

àL 151-152 the average macrobenthos biomass including R. philippinarum was 5,990 ± 3,837 g/m2 (Figure. 2)

  1. L181. Please change the word "abundance" to "density" for the sake of consistency. The same goes to the column labels so they read "average density."

à As your comments, we changed the word “abundance” to “density”.

àL 180-181 The average density (fourth-root transformed data) and contribution (%) to the dissimilarity of each species are shown,

6 L194. "Additionally, macrobenthos biomass was positively correlated with R. philippinarum biomass." Again, please clarify whether R. philippinarum biomass was included in the calculation of macrobenthos biomass in the text. This is important, as if it was NOT included, it means that higher biomass of R. philippinarum is associated with higher biomass of other organisms, which does not seem to make sense given the negative correlation between R. philippinarm and Shannon index, but I guess it can happen... If it was included, it probably simply means that a relatively large proportion of the macrobenthos biomass is the biomass of R. philippinarum.

à As your comments, we changed the sentence to “Additionally, macrobenthos biomass including R. philippinarum was positively correlated with R. philippinarum biomass”.

àL 194-195 Additionally, macrobenthos biomass including R. philippinarum was positively correlated with R. philippinarum biomass.

  1. L201-202. "The use of univariate measuresof diversity (Shannon-Weaver diversity; H’) to evaluate the level of stress-induced on macrobenthoscommunities [44,45]." This is not a complete sentence. Please rephrase.

à As your comments, we changed the sentence to “The univariate measures of diversity (Shannon-Weaver Diversity; H ') is used to assess the level of stress in the macrobenthos community.”.

àL 202-203 The univariate measures of diversity (Shannon-Weaver Diversity; H ') is used to assess the level of stress in the macrobenthos community [44,45].

We believe the manuscript has been improved satisfactorily and hope it will be revised for publication in Special Issue “Harmful organisms and their management for sustainable environment” of Sustainability.

Best regards,

OK HWAN YU

This manuscript is a resubmission of an earlier submission. The following is a list of the peer review reports and author responses from that submission.

Round 1

Reviewer 1 Report

Although the topic of the manuscript is quite interesting, I find many incostintencies and serious problems with sampling design.  

The Introduction is fairly good but when I went through M&M I noticed something really strange. I understand the authors sampled only one station with two replicates (not even three!) for 4 years? This is not acceptable.

Results are very messy, badly described and very often AA refer to wrong figures and tables.

Another major point is related to the elimination of Ruditapes from the statistical analysis of macrobenthos, without giving any explanation for that. PERMANOVA is not appropriately explained  and is also incomplete. 

In the light of these comments, unfortunately, my opinion is to reject the present manuscript.

However, I enclose a pdf file with my comments throughout the draft.

Reviewer 2 Report

The ms by Kim et al. on The Influence of Manila Clam (Ruditapes Philippinarum) on Macrobenthos Communities in a Korean Tidal Ecosystem is clearly written and illustrated. I only have few remarks that may be used to improve the ms.

L3. Philippinarum should not be written with a capital: Ruditapes philippinarum

L22. Instead of “affects” you could say “positively affects” for clarity?

L33-34. Can you tell whether the Manila clam has been introduced to Korea or whether Korea is part of its native range?

L45. What is “high”? It may be better is you say “too high”

L58-59. “The distributions of intertidal macrobenthic taxa are strongly influenced by environmental and biological factors.” This is true for most species. Maybe you should tell about which factors are more relevant for intertidal organisms.

L77. Please give coordinates for all localities.

Table 2. I sugest that you name which phylum the species belongs to (between brackets): Crustacea, Mollusca, Polychaet, etc. Most readers are not familiar withe the species names and perhaps mentioning the phylum explains more about the relation between this species and the Manila clam (competitor, associated species, etc).

L206. Many factors, such as tidal amplitude and currents ?

Reviewer 3 Report

This study describes macrobenthic communities in a tidal ecosystem in Korea.

I have a few major concerns about this study.

First, it is not very clear what primary goals of this study were. If it is to investigate the influence of Manila clams on macrobenthic communities (as the title indicate), it makes much more sense to have more spatial coverage, collecting samples from similar tidal environments with different levels of clam density.  Instead, authors used a single (?) site and conducted repeated sampling.  Repeated sampling is generally done to investigate any temporal changes in the community, so the sampling design doesn't seem to be appropriate.

Another concern is the low (?) number of replicate samples.  It sounds like (from lines 76-77) 2 replicate samples were collected each time.  The method section sounds like there were 4 different sites (though the map only shows 1 site), so were there 8 samples each time?  Looking at the figures 4&5, it also looks like authors pooled all samples for each sampling period, so again, not very clear how many replicates were collected.  Looking at temporal changes in biological communities and environmental factors, there seems to be lots of temporal variability in the dataset, so this may be the situation more replicate samples should be used.  Authors also do not explain or try to speculate what the causes of the observed temporal variability (e.g. changes in grain size or species richness/diversity).

Finally, some of the statistical analyses do not seem to be done appropriately.  More specific comments are listed below, but if authors had replicate samples, analyses should have been done on those replicates, not data pooled per sampling period.

I recommend doing extensive work to specify clear goals and research questions and analyze data appropriately in a way that can clearly answer the questions.  More specific comments are listed below.

Introduction

L59. Please add some examples of "environmental and biological factors".

L62-69. Please elaborate as the paragraph is currently too vague.  Some examples (brief descriptions) of earlier important studies, as well as brief explanation of interference vs. exploitation competition would be helpful.

L70-73. Please expand this paragraph by adding the goal(s) of the study, potential importance of the study to ecology and resource management, etc. It is also odd that the paragraph prior to this talks about competition, yet you don't mention anything about it here.  How does competition relate to your study?

Materials and Methods

L76. What do you mean by "seasonally"?  4 time a year?  Please specify in the text.

L77. Please add reference to Figure 1. It also seems figure and table numbers are off in the text.

L80. Please justify why you only collected 0-1 cm layer for grain size analysis when your biological samples came from 0-30 cm depths.

L97. Please explain why clams were removed from your analyses.  Was it all clams or just Manila clams?

L100-101. Please rewrite this sentence.  Based on my experience, I assume you used the Bray-Curtis similarity measure calculated on 4th-root transformed density data and used group-average linkage for the cluster analysis, but that's not how the sentence reads.

L105. Please rewrite this sentence.  POC will not let you investigate "the distance between benthic organisms."  What do you mean by the distance between benthic organisms, anyway?

L107. If PERMANOVA was applied to any univariate data in your study, please add which variables were analyzed and which distance measure was used (Bray-Curtis wouldn't be appropriate).  If not, the statement "This test can be applied to both multivariate and univariate data" is not really relevant here (though it is true).

L109-113. Looking at Figure 5, this sentence does not describe what you actually did.  PCO here is an unconstrained ordination using the first 2 PCO axes that will let your samples group based on similarity/dissimilarity measured on the basis of Bray-Curtis.  You are then looking at correlations, through vector overlay function of PRIMER, between the PCO scores and those factors (TOC, RP densities etc.) to explore variables that potentially contributed to the observed pattern (grouping of samples). Vector overlay is an exploratory tool and all it shows is a correlation.  The variables identified using this tool may or may not explain anything.

Results

Table 1. I suggest converting this to a plot showing the mean grain size and TOC.  Sediment types can be listed on the top or you can use different symbols for grain size to show different sediment type.  Please also include what each sediment type means in the caption.

L142. "although particularly low diversity was observed in April 2014 (1.62)". I don't see this in the figure.

L155. Is the three grouping based on a permutation (SIMPROF) test?  If so, please note the significance level (α).

L160-161. Please note your PERMANOVA design in the methods. (e.g. 2-way nested, with both year and season factors random, etc).  Without this information, there is no way to tell if this analysis was appropriate and meaningful, and actually looking at Table 3, I do not think the analysis was appropriate.  PERMANOVA analysis should be planned a priori.  Here, it seems like you used "Cluster" grouping as a factor, which was already identified by another statistical text, namely SIMPROF test (though this is not very clear in the text), to be significantly different from one another.  All this test shows is that SIMPROF and PERMANOVA yield consistent results.  What would have been appropriate and potentially meaningful was to test for any year-to-year and/or seasonal variability in the data, so using Year and Season as factors, though this depends on your hypothesis.

Figure 4. Assuming the grouping of cluster analysis reported in the text was based on SIMPROF test, there is not really a point of showing both cluster and MDS.  I recommend showing just MDS with cluster overlay with SIMPROF results. The MDS plot currently looks like the envelop was drawn at 52% similarity and it is not clear whether this number was based on SIMPROF test or simply eyeballing the cluster tree and picking it arbitrarily out of convenience (which is not appropriate). I assume it was based on SIMPROF, but it's should be clarified in the figure/text.

L174-177. Given that the variability in the biological community explained by the first 2 PCO axes is relatively low and the very fact you do formally investigate the relationship between the biological community and environmental variables using BIO-ENV, there is no point of doing PCO analysis.

Discussion

L200. This study did not compare the study area with the surrounding areas, so how can you say "the study area is less healthy than surrounding areas"?  It is also not clear what you mean by "healthy".

L212. What you found are correlations, so you cannot say Mz, TOC and R.philippinarum biomass influenced clustering. A causal relationship was not established.